# The Tongue in Three Species of Lemurs: Flower and Nectar Feeding Adaptations

**DOI:** 10.3390/ani11102811

**Published:** 2021-09-27

**Authors:** Juan Francisco Pastor, Magdalena Natalia Muchlinski, Josep Maria Potau, Aroa Casado, Yolanda García-Mesa, Jose Antonio Vega, Roberto Cabo

**Affiliations:** 1“Osteology and Compared Anatomy” Research Group, Departament of Anatomy and Radiology, University of Valladolid, 47005 Valladolid, Spain; juanpas@med.uva.es; 2Anatomical Sciences & Education Center, Oregon Health and Science University, Portland, OR 97239, USA; muchlins@ohsu.edu; 3Unit of Human Anatomy and Embryology, University of Barcelona, 08036 Barcelona, Spain; jpotau@ub.edu (J.M.P.); aroa.casado@ub.edu (A.C.); 4SINPOS Research Group, Departament of Morphology and Cell Biology, University of Oviedo, 33006 Oviedo, Spain; garciamyolanda@uniovi.es (Y.G.-M.); javega@uniovi.es (J.A.V.); 5Faculty of Health Sciences, Autonomous University of Chile, Santiago 8380447, Chile

**Keywords:** *Lemur catta*, *Varecia variegatta*, *Eulemur macaco*, Madagascar, sublingua, papillae, ecology, coevolution, Chievitz

## Abstract

**Simple Summary:**

Comparative studies of primate gustatory anatomy are sparse and incomplete. Here, we compare both the macro and micro anatomy of the tongue in three lemurid species. We included two non-destructive nectar-feeders, *Varecia variegata* and *Eulemur macaco*, and one destructive flower feeder, *Lemur catta*. To study the tongue’s structure, we used direct observation together with different microscopy techniques, ranging from optical microscopy to electronic microscopy. We found differences in the size, shape, and distribution of the tongue’s papillae. Most notably, there are large distinct papillae present at the tip of the tongue in nectar-feeding species. In addition, histological images of the ventro-apical portion of the tongue show that tongue houses an encapsulated structure in all species studied. The non-destructive flower-feeding species share similar tongue and sublingua anatomy, suggesting that the observed features may be adaptive for flower feeding. These features were not observed in the destructive flower feeder, *Lemur catta*.

**Abstract:**

The mobility of the primate tongue allows for the manipulation of food, but, in addition, houses both general sensory afferents and special sensory end organs. Taste buds can be found across the tongue, but the ones found within the fungiform papillae on the anterior two thirds of the tongue are the first gustatory structures to come into contact with food, and are critical in making food ingestion decisions. Comparative studies of both the macro and micro anatomy in primates are sparse and incomplete, yet there is evidence that gustatory adaptation exists in several primate taxa. One is the distally feathered tongues observed in non-destructive nectar feeders, such as *Eulemur rubriventer*. We compare both the macro and micro anatomy of three lemurid species who died of natural causes in captivity. We included the following two non-destructive nectar feeders: *Varecia variegata* and *Eulemur macaco*, and the following destructive flower feeder: *Lemur catta*. Strepsirrhines and tarsiers are unique among primates, because they possess a sublingua, which is an anatomical structure that is located below the tongue. We include a microanatomical description of both the tongue and sublingua, which were accomplished using hematoxylin–eosin and Masson trichrome stains, and scanning electron microscopy. We found differences in the size, shape, and distribution of fungiform papillae, and differences in the morphology of conical papillae surrounding the circumvallate ones in all three species. Most notably, large distinct papillae were present at the tip of the tongue in nectar-feeding species. In addition, histological images of the ventro-apical portion of the tongue displayed that it houses an encapsulated structure, but only in *Lemur catta* case such structure presents cartilage inside. The presence of an encapsulated structure, coupled with the shared morphological traits associated with the sublingua and the tongue tip in *Varecia variegata* and *Eulemur macaco*, point to possible feeding adaptations that facilitate non-destructive flower feeding in these two lemurids.

## 1. Introduction

### 1.1. Nectar Feeding

Coevolution has produced a codependency between primates and plants [1,2,3,4]. Primate nectar feeding and the cross-pollination of flowers is well acknowledged today [5,6,7,8]. Among extant primates, the best evidence for non-destructive flower feeding exists among the Malagasy strepsirrhines. Nectar appears to be the second most important food item in the annual diets of most Malagasy strepsirrhines [9], and can sustain a medium-sized primate by providing them with protein, sugars, and essential amino acids [10,11,12,13,14,15]. Malagasy strepsirrhines that regularly or seasonally feed on nectar are *Cheirogaleus major*, *Microcebus rufus* [16], *Eulemur mongoz* [17,18,19], *Eulemur rubriventer* [9,20], *Eulemur fulvus rufus* [9,14], *Eulemur fulvus sanfordi*, *Eulemur coronatus* [9], *Eulemur macaco* [9,20,21,22], and *Varecia variegata variegata* [12,13]. *Eulemur mongoz mongoz* has been observed to spend 80% of its foraging time feeding from the nectar-producing parts of four species of plants [5,18,23]. Eighty percent of this time was spent on the nectar of the kapok tree (*Ceiba pentandra*) [13,23]. Similar feeding behaviors are observed year-round for *Eulemur mongoz*. *Varecia variegata variegata* spends approximately 25% of its feeding time exploiting nectar, making nectar the second most important food item in its diet [12,13]. However, during periods of fruit scarcity, the Traveler’s Palm (*Ravenala madagascariensis*) is in bloom. During this period, over 72% of *Varecia*’s feeding time is spent on this one plant; thus, Ravenala is *Varecia*’s primary source of caloric intake [12]. It is important to note that neither *Eulemur mongoz* nor *Varecia variegata* are destructive to any part of the plant. Non-destructive nectar feeding, when combined with the morphology of some “lemur-loving” plants (e.g., large durable flowers) [12,13], implies a possible mutualism between certain Malagasy strepsirrhines and Malagasy plants. The circumstantial evidence suggests that many Malagasy plant species may actually depend on animals as important cross-pollinators [12,13,17,23,24].

### 1.2. The Tongue in Prosimian Primates

The tongue is a highly mobile organ that occupies the lower half of the oral cavity. The mobility of extrinsic and intrinsic tongue muscles, combined with the rich innervation of the overlying mucosa, allows the tongue to not only reach and explore potential food resources, but it is central in formulating informed nutritional decisions. Across chordates, the tongue has been adapted and modified to carry out particular forms of feeding [25,26,27]. Among prosimian primates (i.e., lemurs, lorises, galagos, and tarsiers), variation in tongue morphology has been linked to dietary adaptations, most notably, nectar feeding [14,28]. However, what is truly unique among prosimian primates is the presence of a sublingua, a second tongue-like structure that is located below the primary tongue. The sublingua is purported to assist with the dental hygiene of the so-called “tooth comb”, which consists of the horizontally oriented mandibular incisors and canines [29,30]. The gross anatomy of the tongue has been fairly well documented among primates, but very little is known about tongue microanatomy.

### 1.3. Gross Anatomy of Tongue

The tongue is divided in the anterior two-thirds, the body, and the posterior third, the root (often referred to as the pharyngeal tongue). It is composed of a series of intrinsic muscles and extrinsic muscles. The extrinsic muscles originate outside the tongue (e.g., styloid process of the temporal bone), and are used to protrude, retract, and laterally deviate the tongue, while the intrinsic muscle modifies the shape of the tongue. The intrinsic tongue musculature, located on the dorsum of the tongue, is covered by a topographic mucous membrane. The topographic appearance is due to the presence of a vast number of papillae, sorted as gustatory and non-gustatory. It is within these gustatory papillae that taste buds are found. Taste buds house a cluster of taste cells that transmit specific tastants to the central nervous system. The papillae can be observed on the dorsum of the tongue. The posterior one-third of the tongue, the root, is often referred to as the pharyngeal tongue. Beyond innervation, the most distinct feature of this region is the large cluster of lymphoid follicles, which, together, are referred to as the lingual tonsils. Although human anatomy [31] provided the supporting anatomical statements, similar features are observed among the lemurids, to the exclusion of the following: (1) no sulcus terminalis, and (2) a “Y-shaped” arrangement of the circumvallate papillae at the root of the tongue [32,33].

### 1.4. The Tongue’s Papillae

Papillae are raised regions within the mucosa of the soft palate, uvula, epiglottis, pharynx, larynx, esophagus, and tongue [34]. There are the following three types of gustatory papillae on the dorsal surface of the primate tongue: circumvallate, foliate, and fungiform [28,35]. Filiform papillae are non-gustatory papillae, which carpet the dorsum of the tongue. Conical papillae, which carpet the posterior third of the tongue, are considered, for us, as another type of non-gustatory papillae, whereas others considered them to be a subtype of filiform papillae [36]. The gustatory papillae are distinct in where they are located on the tongue, and they each have distinguishing features [37]. Circumvallate are located on the posterior aspect of the tongue, surrounded by conical papillae, whereas foliate are located on the posterolateral aspect of the tongue. Fungiform papillae are located on the anterior two-thirds of the tongue and are often the first gustatory structures that come into contact with an ingested chemical substance, and are critical for food selection [34,37].

### 1.5. Aim of the Work

The comparative histomorphology and microanatomy of the tongue and sublingua of the following three Malagasy prosimians are examined here: *Varecia variegata variegata* (the black and white ruffed lemur), *Eulemur macaco* (the black lemur), and *Lemur catta* (the ring-tailed lemur). These species were selected, partially because of their reported feeding ecology. *Varecia* and *Eulemur* are specialized non-destructive flower and nectar feeders, while *Lemur* is more generalized in its feeding preferences and has not been documented to be a non-destructive flower feeder. Alport [35] provided one of the first comparative studies into fungiform papillae. Fungiform papillae appear to be found in all the primates sampled [35]. However, many strepsirhines appear to lack foliate papillae and have fewer circumvallate papillae [28,38]. One of the primary objectives of this study is to document the topographic anatomy of the dorsum of the tongue. In a recent review, Iwasaki [28] listed *Lemur* and *Eulemur* as strepsirrhine species that lack foliate papillae, and nothing has been described for the nectar-feeding *Varecia*. We have carried out detailed histological and microanatomical examinations, with the aim of (1) completing the current knowledge of the morphology and location of the papillae on the tongue; (2) bringing useful data to light, to aid in taxonomical tasks; and (3) contributing to a better understanding of anatomical patterns, as they relate to diet and ecology.

## 2. Materials and Methods

### 2.1. Specimen Acquisition and Preparation

The tongue and sublingua were removed from seven *Lemur catta* (2 females and 5 males), five *Varecia variegata* (2 females and 3 males) and two *Eulemur macaco* (a female and a male) specimens. The specimens were photographed with a Nikon digital camera COOLPIX D-3000. The lemurs were obtained through donations of naturally dead specimens from Spanish zoos (Zoo Santillana del Mar, Bioparc Fuengirola and Bioparc Valencia) where the animals were fed with special commercial diets for lemurs, supplemented with fresh fruits. Immediately upon death, the lemurids were frozen at their respective zoos. They remained frozen upon arrival to the Anatomy and Radiology Department, Anatomical Museum, at the Universidad de Valladolid, where they were again placed in freezer to maintain specimen integrity. To obtain the tongues, the animals were allowed to thaw for 24 h in a refrigerator. The jaw was lowered, and the tongue was pulled with forceps. Once the tongue had been placed in antepulsion, it was sectioned at the level of the glosso-epiglottic fissure (vallecula) using a scalpel. The samples obtained were immersed in 10% formaldehyde in a glass jar marked with data corresponding to species, sex and date, and kept in that way until their use. The animal was frozen again for future research.

### 2.2. Macroscopically Observation

Together with direct observation we used a Leica M205FA fluorescence stereo microscope. Semi-automatic stereo microscope for transmission (contrast and relief), reflection and fluorescence studies. Optics: 1× (working distance: 61.5 mm), 2× (working distance: 20 mm) and 5× (working distance: 19 mm). Zoomed from 0.7 to 20.5×. Reflection light by LED ring with 1/4, 1/2 or full-ring illumination. Digital color camera, Leica DFC310FX, maximum resolution 1392 × 1040 pixels (1.4 Mpixels CCD). Scientific-technical services, Universidad de Oviedo.

### 2.3. Fixation and Processing for Histological General Stains (Hematoxylin–Eosin and Masson Trichrome)

Fixation was performed with 10% formaldehyde in 0.1 M phosphate buffer (room temperature, pH 7.2) for at least 24 h in agitation. Thereafter the pieces were dehydrated and processed for routinely embedding in paraffin. The paraffin-embedded samples were cut into 10-µm-thick sections parallel to the tongue’s frontal or sagittal plane and mounted on gelatin-coated microscope slides. The structure of the tongue samples was assessed using Mayer’s hematoxylin–eosin and Masson trichrome stain (Kit de Masson’s trichrome, Poly Scientific R & D Corp., Bay Shore, NY, USA). The stained sections were washed in water, dehydrated in an ethyl alcohol battery in increasing concentrations, diaphased in xylol and mounted with Entellan^®^. The preparations were photographed under a Nikon Eclipse 80i light microscope attached to a Nokia DS-5M camera.

### 2.4. Single Immunohistochemistry

Deparaffinized and rehydrated sections were processed for detection using the EnVision antibody complex detection kit (Dako, Copenhagen, Denmark), following supplier’s instructions. Briefly, the non-specific binding was blocked with 50% bovine serum albumin in TBS for 20 min. Sections were then incubated overnight at 4 °C with the primary antibody (rabbit anti-cow S100 protein, code n° Z0311, Dako; cross-reaction with human S100A and B protein, not tested in lemurs). Subsequently, the sections were incubated with the anti-rabbit EnVision system-labelled polymer (Dako-Cytomation) for 30 min, washed in buffer solution, and treated with peroxidase blocking buffer (Dako Cytomation). Finally, the slides were washed with buffer solution and the immunoreaction was visualized with diaminobenzidine as a chromogen, washed, dehydrated, and mounted with Entellan^®^ (Merk, Dramstadt, Germany).

### 2.5. Ultrastructural Study Using Scanning Electron Microscopy (SEM)

This study was performed on the dorsal surface of the tongue. We used scanning electron microscopy and environmental scanning electron microscopy (the same protocol, but without metallizing the samples). The samples were immersed in a solution of hydrochloric acid 1 N concentrated, at room temperature, ranging from 10 to 20 min (depending on the sample size, less size less time) in agitation. This treatment was performed to remove debris from sloughed cells. Then, after several steps of rinsing in tap water, the pieces were dehydrated in a graded acetone series, critical-point dried in a Balzers CPD 030, sputter coated with 3 nm gold in a Balzers BAL-TEC SCD 050 and examined under a conventional scanning electron microscope MEB JEOL-6100 (Scientific-technical services, Universidad de Oviedo) or with an ESEM Quanta 200 FEG scanning electron microscope (Scientific Park Foundation, Universidad de Valladolid).

## 3. Results

### 3.1. The Tongue

#### 3.1.1. General Observation

As in all primates, the shape of the tongue is elongated, with a rounded tip. The lateral borders are parallel in *Lemur catta*, but slightly convergent in the forward direction in *Varecia variegata* and *Eulemur macaco*. *Varecia variegata* present regular transverse folds that are visible on the dorsal surface (Figure 1E). The posterior third is in connection with the floor of the mouth, and is where the muscles, vessels, and nerves penetrate. No intermolar prominence or marginal folds are observed on the lingual dorsum. On the lower surface, a well-developed sublingua is observed. In *Lemur catta*, the sublingua is triangular in shape and ends in a single tip; while several tips were observed in *Varecia variegata*. The sublingua of *Eulemur macaco* presents as two asymmetric lobes (Figure 2). On the dorsum of the tongue, the following five types of papillae are distinguished: filiform and fungiform on the body, foliate on the edges at the level of limit between the body and root, and circumvallate and conical on the root of the tongue (Figure 1). Figure 3 shows the ventral part of the tongue and the sublingua. Histological sections of the root of the tongue show muscular fibers (intrinsic and extrinsic muscles) surrounded by abundant adipose tissue, to provide energy and serve as a sliding surface between different muscle bundles (Figure 4). We found a longitudinal structure (Figure 5) inside the apical part of the tongue, such structure appears to be encapsulated and is present in all the species studied (Figure 4B,E,H, framed in Figure 4A,D,G and Figure 5). The inner part is filled with adipose tissue and nerve fibers (both positive for anti-S100 protein, Figure 5E), some muscle fibers and small vessels (capillaries), and connective tissue. Only in the *Lemur catta* specimens was document to have cartilaginous tissue (Figure 4B, faint positive for anti-S100 protein in Figure 5E). Although the antibody to detect the S100 protein (the protein found in myelinic nerve fibers, adipocytes, and cartilage; Figure 5E) has not been tested in lemurs, this antibody appears to signal the presence of similar tissues types in the lemurs to those observed in humans (Figure 5E). In addition to the presence of an encapsulated structure, there appears to be a longitudinally oriented arteriole located within it. A full description of the notable tissue/cell types can be reviewed in Figure 5.

#### 3.1.2. The Tongue’s Tip

The direct and SEM observations showed that *Varecia variegata* and *Eulemur macaco* present, on the tip of the tongue, papillae with an increased size and a modified morphology (Figure 3). Papillae from *Varecia variegata* are hammerhead or mace-like shaped (Figure 7K,L), while those in *Eulemur macaco* seem to be big flask-shaped formations, ended in a “filiform apex” that is forwardly oriented (Figure 3). Conversely, the papillae found on the tongue of *Lemur catta* are uniformly distributed on the anterior two-thirds of the tongue, but the morphology of the papillae are more diverse, ranging from racket-like to conical, and are variable in the number of tips present (Figure 3C). *Lemur catta* did present some papillae that could be described as hammerhead in appearance, on the edges of the tongue, but not on the tip (Figure 4C arrow).

#### 3.1.3. Papillae Location and Morphology

Filiform papillae are located across the anterior two-thirds of the dorsum of the tongue, and reach about 3 mm at the bottom of the tip, along the ventral surface. Its shape is elongated and finished in several points that are directed upwards or backwards. They are highly keratinized and the inner core can be observed with Masson trichrome stain (Figure 6F, Figure 7B and Figure 8B).

Fungiform papillae are located over the entire surface, although they are scarce in the posterior third. These papillae are also located along the entire edge of the tongue. The apex of the fungiform papillae is rounded and the base presents a cylindrical shape. Masson trichrome allowed us to display that the inner part of the fungiform papillae presents a cactus-like shape in the connective tissue in the three species (Figure 6F, Figure 7B and Figure 8B). In the case of *Lemur catta*, the fungiform papillae on the dorsum of the tongue decreased in size from the posterior to anterior (Figure 1A). We found hammerhead or mace-shaped fungiform papillae that could be differentiated from traditional “mushroom”-shaped fungiform papillae, and were present on the edges of the *Lemur catta* tongue (Figure 4C, arrow) and on the tip of the *Varecia* tongue (Figure 7K,L).

Circumvallate papillae are located in the posterior third of the tongue. They are rounded or polygonal shaped. These papillae have a differentiated central portion and are affiliated with an elevated groove. If you were to connect the circumvallate papillae, the letter “Y” would appear. The inferior limb of the “Y” sits posterior between the conical papillae (Figure 6I, Figure 7F and Figure 8D). The number of circumvallate papillae is variable, and there can be between four and seven papillae, six was the typical number (a couple in each branch of the “Y”). Its structure is composed of a central papilla, escorted around by papillae with a more or less modified shape, forming a vallum-like perimeter. In the case of *Lemur catta* (Figure 1F and Figure 6D,E), the circumvallate papillae are surrounded by conventional conical papillae, whereas in *Varecia variegata*, the conical papillae present morphological changes that consisted in the presence of several tips that mimic a cockscomb (Figure 1F and Figure 7D,H). Note the presence of taste buds within the upper margins of the circumvallate papillae, in addition to the ones on the lateral side (Figure 7E). In the Eulemur macaque specimens there seem to be true circumvalate papillae, with a central part mainly surrounded by papillae without tip or crest, and even in the posterior branch of the arrangement of these papillae in the form of a “Y” it presents pairs of papillae of this type surrounded by a “common vallum” (Figure 8D,F).

Conical papillae are distributed on the posterior third of the tongue. In *Lemur catta*, their tip is simple (Figure 1B and Figure 6C,D). In *Varecia variegata*, those surrounding the circumvallate papillae present several tips, similar to a cockscomb, and the rest present a single tip. *Eulemur macaco* present a single tip, except those surrounding the circumvallate papillae that, in the majority, have lost the tip. They are lying down, directing their tips backwards.

Foliated papillae are located on the edges of the zone that limits the transition between the body and the root of the tongue, and several have folds that are separated by grooves. The folds and grooves are arranged perpendicularly to the horizontal plane of the tongue. They are present in all the specimens studied (Figure 1C,G,K), including *Eulemur macaco* (Figure 1K and Figure 8G–I), which present very narrow grooves, which make identification difficult.

### 3.2. The Sublingua

The macroscopic observation showed the typical triangular, arrowhead or heart-shaped morphology (Figure 2A,D,G). The free anterior part ends in a single narrow tip in *Lemur catta* (Figure 2B,C); whereas, in the other two species, it finished in a wide tip (in Figure 2, it is like a brush in the case of *Varecia variegata* (Figure 2E,F).

## 4. Discussion

Formerly, evolutionary variations in vertebrate tongue morphology have been linked to habitat adaptations [38]. Among strepsirrhine primates (e.g., lemurs, lorises, and galagos), variation in tongue morphology has been linked to dietary adaptations, most notably, nectar feeding in Malagasy lemurids [14,28]. There are few in-depth morphological studies that complement the numerous existing ones on behavior. Studies on the tongue of lemurs are scarce; some are from the last century and some did not have high-resolution techniques at their disposal, or included few specimens per species [14,33,39,40,41,42]. Here, we examined seven *Lemur catta*, five *Varecia variegata*, and two *Eulemur macaco* specimens using a wide range of imaging techniques, including direct observation, and optical and electronic microscopy, to document the different lingual papillae and discuss how diet could have affected the observable anatomical variation [39].

Feeding decisions in primates are guided by the information provided by the gustatory papillae. Thus, the distribution of papillae on the tip of the tongue, the overall tongue mobility and length, as well as microanatomical features of the tongue can help a primate to be more successful in acquiring a food resource. Nectar-producing flowers vary in color, shape, size, and overall anatomical construction. It appears that nectivorous animals frequently consume different types of flowers depending on their body size [8]. Many flowers have superficial nectaries, meaning that nectar is produced close to the entrance of the corolla, while others have deeper nectar chambers. Large-bodied nectar feeders, such as *Varecia* and *Eulemur*, tend to exploit flowers that are large and sturdy in construction, and have deep nectaries [12,13,21]. Similar to a lock and key, the head of *Varecia* is closely matched in shape to *Ravenala madagascariensis*, the Traveler’s Palm [8]. The nectar from this plant’s flower can represent over 72% of *Varecia*’s diet [12,13]. The elongation of the muzzle and cranium that is observed in nectar feeders has been reported to assist these animals in reaching the base of a flower, where the nectar is located [8]. When the head is withdrawn from the deep corolla of the flower, pollen collects on the muzzle and head. A non-destructive nectar feeder moves pollen from flower to flower, removing nutrient-rich nectar rewards and leaving pollen behind, probably facilitating cross-pollination and strengthening co-evolutionary bonds.

The long muzzles of *Varecia* and *Eulemur* do not only assist in reaching deep nectar chambers non-destructively, but they also house the tongue. Nectivores are described as having long “prehensile” tongues that often have modified papillae [43]. Descriptive terms, such as “feathered” or “brushed”, are also used to describe features of the hair and tongue of nectar-feeding mammals [14,25,26,27,43]. Similar features were observed among the two non-destructive nectar feeders in our sample. For example, Figure 1D,H,L compared the gross anatomical differences in the density of the fungiform papillae, relative to the shape/texture of the filiform papillae, across the three lemurids. These images clearly show cockscomb-like or feather-like filiform papillae in the non-destructive nectar feeders (Figure 1H,L), and smooth filiform papillae of *Lemur catta* (Figure 1D). Upon even closer scrutiny (Figure 7 and Figure 8), the morphological patterns affiliated with non-destructive nectar feeding continue. *Eulemur macaco* has large flask-like-shaped filiform papillae on the tip of the tongue. All these apical papillae form a kind of feather or brush on the tongue’s tip, similar to that described by Overdorff [14] in *Eulemur rubriventer*, which is another non-destructive flower feeder. In addition, we observed conical-like papillae with one or more tips.

Nectar feeders are rewarded with a nutrient-rich liquid, containing proteins, sugars, and seven essential amino acids, if they feed non-destructively. The fungiform papillae are the first gustatory papillae that come into contact with a tastant and can be more sensitive to sweet tastants [35]. We found that *Varecia* has a large hammerhead-shaped papillae (with many taste buds and pores) on the tip of the tongue. The hammerhead papilla seems to be a modified form of fungiform papillae, and this modification may provide a wider sensory (gustatory) detection surface on the tip of the tongue. The presence of modified fungiform papillae has been described in humans [44].

Beyond the modified apical fungiform papillae, the overall histomorphology of the fungiform papillae among our captive lemurid population is in line with the reported observations by Takemurao [41] and other comparative primate studies [45]. Using Masson trichrome stain, we were able to confirm that fungiform papillae have a “cactus-shaped” connective tissue center, which is a feature similar to those observed in several species of insectivores and lagomorphs. It must also be taken into account that, due to the few specimens available for our study, as reported by Yoshimura [45] in his study on *Colobus guereza*, the internal structure of the papillae varies with age, presenting, in the inner core, a shape that is more similar to a cactus in the most senescent than that in young specimens.

Regarding the foliate papillae, in former studies of the *Eulemur fulvus rufus* [14,40] and *Eulemur rubriventer* [14] tongue, the presence of filiform, fungiform, and circumvallate papillae were cited, but there was no mention of foliate papillae. Along these lines, in Shin-ichi Iwasaki’s [28] review of the primate tongue, there was no discussion of foliate papillae, and it was stated that they were not found. In the two specimens of *Eulemur macaco* studied, we found these papillae located laterally on the tongue, just as they are described in humans. Although not systematically documented by Alport’s [35], similar observations were reported about the presence and the location foliate papillae. We suspect that foliate papillae are found in all primate species, but because these papillae lack strong pigmentation, coupled with the gustatory crypts where the taste buds are found, these papillae are hard to see. In fact, it was only high-magnification microscopy and SEM that allowed us to identify these papillae within *Eulemur macaco*.

One of the most interesting observations made in this study, from a morphological perspective, was the comparative analysis of circumvallate papillae. These papillae are named after a particular structure, the “vallum” (*Latin* for wall), located within their chemosensory area of the posterior one-third of the tongue. The vallum limits and isolates the taste buds and pores. The taste buds/pores are located on the lateral surface of the central chemosensory papilla, which is located medial to the vallum. Just as with folate papillae, the taste buds are found within crypts—allowing tastants to pool within them, eliciting a gustatory response. Sonntag [33] studied several species of lemurs, and described them as “… round or oval, and are surrounded by an annular or lobulated vallum”. We found that the circumvallate papillae in the three species presented differently—not only in anatomy, but in how they are organized/arranged. In *Lemur catta*, tastants are directed to the gustatory crypt surrounding the isolated papilla, by the undifferentiated conical papillae surrounding them. In *Varecia*, the vallate papillae are surrounded by flattened conical papillae and have several crests. In *Eulemur*, one (those forming the lateral branches of the “Y-shaped” distribution) or two (those forming the posterior branch) vallate papillae are surrounded by a common structure that is similar to the “vallum” mentioned by Sonntag [33]. There are different types of structures that mimic the “vallum” present in other primates [46]. Demonstrating that there are slight differences in the morphology of circumvallate papilla, and of the structures that surround them (formed with, or derived from, conical papillae, a mechanical papillae), can (1) demonstrate how different functional “vallums” can form; (2) contribute to a better understanding of how the circumvallate papillae function; and (3) could potentially be useful in taxonomic classifications with additional comparative studies.

Nectar feeders are reported to have exceptionally mobile tongues [43]. We found an encapsulated structure in the antero-ventral position of the tongue, which may help with tongue prehensility and mobility. Iwasaki et al. [28] state that the tongues of strepsirrhines are more precise and better equipped than haplorhines to make fine movements outside the oral cavity. Esta estructura encapsulada, ubicada en la parte anteroventral de la lengua, se asemeja a otra estructura encapsulada como es el órgano yuxta-oral de Chievitz (JOO, for review see [47]). The JOO are important to control movements during oral cavity tasks, such as mastication, swallowing, and sucking [48]. In addition to morphological and functional similarities, the part of the tongue that bears this structure develops from a similar region, the first pharyngeal arch, just as the human JOO does [49]. We think that this particular structure, owing to its key position and morphology, could serve sensory information to contribute to the fine movements of the tip. Moreover, the destructive flower/nectar feeder *Lemur catta* appears to have a cartilaginous structure within this encapsulated organ (as S100 positive immunostaining and morphology shows), which, theoretically, would allow less mobility than the other two species studied. Additional behavioral and anatomical studies should be carried out before we can characterize this structure and substantiate our hypotheses.

For completion, we included a brief comparison of the sublingua in all of the three species studied. We found a narrow tip in *Lemur catta*, in contrast with a wider end for the tips in the nectar-feeding species. Wider and “feathered” tongues have been reported in nectar-feeding bats [43]. Additionally, the sublingua tip of *Varecia variegata* appears to be feathered or brushed, which is a feature observed on the tongue of nectar-feeding animals [9]. The function of the sublingua in lemurs has been associated with “toothcomb” cleaning [50]. Toothcombs are a unique strepsirrhine trait, where the incisors and canines of the lower dental arcade are procumbent (tilt forward). Lemuriformes use the toothcomb and their modified “claw-like” nail on their second hindlimb digit to clean the woolly fur of affiliated conspecifics [30]. Although it is likely that the sublingua does help with cleaning the toothcomb, there is currently not enough evidence to attribute the function of “toothbrush” to it, a view that is shared by other primate anatomists [51]. We performed an ultrastructural study with chemical analysis (unpublished and preliminary) on the ventral surface of the sublingua, and found pores with protruding substances that could be related to other functions, such as digestive collaboration. These digestive functions could facilitate dental cleaning, but could also help break down complex compounds found in pollen or nectar (or other foods). In the absence of detailed comparative microanatomical studies, our observations about the sublingua ultrastructure, and the relation of the sublingua with the tongue/oral cavity, suggest that the sublingua may serve an additional role beyond dental hygiene. The observed morphological diversity in the closely related lemurids species studies indicates that subtle ecological pressures may have shaped the morphology in the two non-destructive nectar feeders (*Varecia* and *Eulemur*). Additional anatomical and, more importantly, behavioral studies of the sublingua are needed to understand the significance of this anatomical trait.

The documented differences found in (1) papillae morphology and location, (2) the histomorphology of the lingual encapsulated structure, and (3) broader sublingua in *Varecia variegatta* and *Eulemur macaco* when compared with *Lemur catta*, point towards the possibility of feeding adaptations associated with non-destructive flower feeding. We hope that the morphological characterization of the papillae in this species could aid in the taxonomical classification of extant lemurids. We also encourage behavioral research teams to work hand-in-hand with laboratories that can undertake ultrastructural studies, so as to offer a broader vision and to characterize, in a better way, the biological legacy that our planet offers us.

## Figures and Tables

**Figure 1 animals-11-02811-f001:**
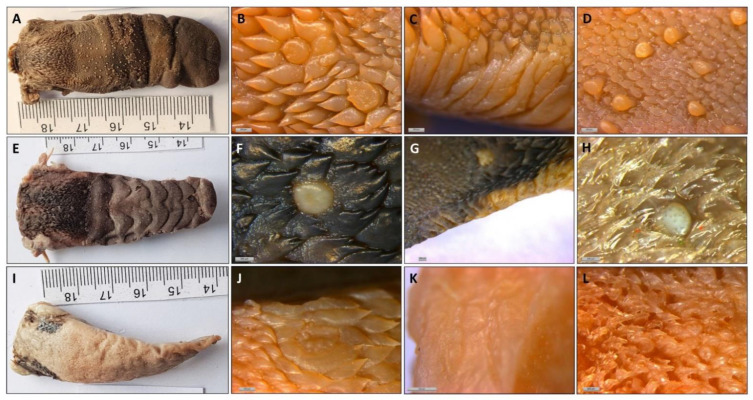
Images from *Lemur catta* (superior row), *Varecia variegata* (middle row) and *Eulemur macaco* (inferior row) showing the dorsum of the tongue (**A**,**E**,**I**) and the zones of the following papillae: circumvallate together with conical (**B**,**F**,**J**), foliate (**C**,**G**,**K**), and fungiform together with filiform (**D**,**H**,**L**). In the left column, the minor division of the scale corresponds to one millimeter.

**Figure 2 animals-11-02811-f002:**
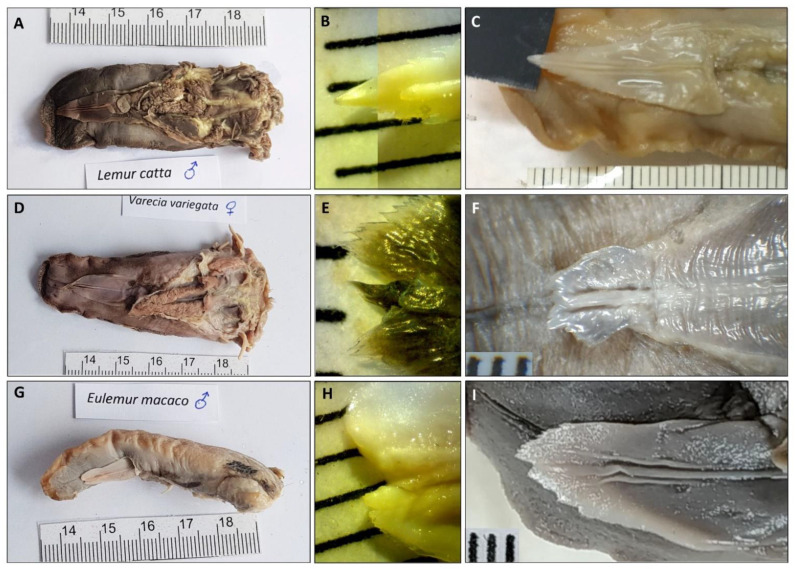
Ventral aspect of tongue and sublingua in Lemur catta (**A**–**C**), *Varecia variegata* (**D**–**F**) and *Eulemur macaco* (**G**–**I**). The sublingua tip (**B**,**C**,**E**,**F**,**H**,**I**) was magnified in several specimens per species. The sublingual tips in the flower-feeding species are wider and more feathered in appearance compared with *Lemur catta*. The minor division of the scale corresponds to one millimeter.

**Figure 3 animals-11-02811-f003:**
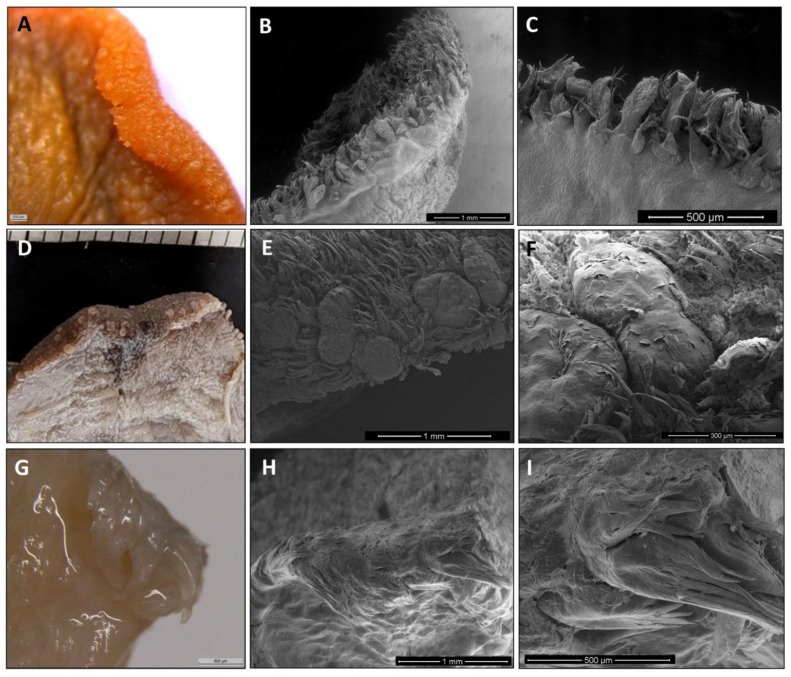
Images from *Lemur catta* (**A**–**C**), *Varecia variegata* (**D**–**F**) and *Eulemur macaco* (**G**–**I**) tongue’s tip. In all images the ventral surface is observed in the inferior part of the picture. The tip in the flower-feeding species present bigger and modified papillae (*Varecia variegata*) or is like a brush with long filiform-like papillae. In *Lemur catta* there are no differences in papillae morphology or distribution when compared with the rest of the tongue’s edge. In image (**D**), the minor division of the scale corresponds to one millimeter.

**Figure 4 animals-11-02811-f004:**
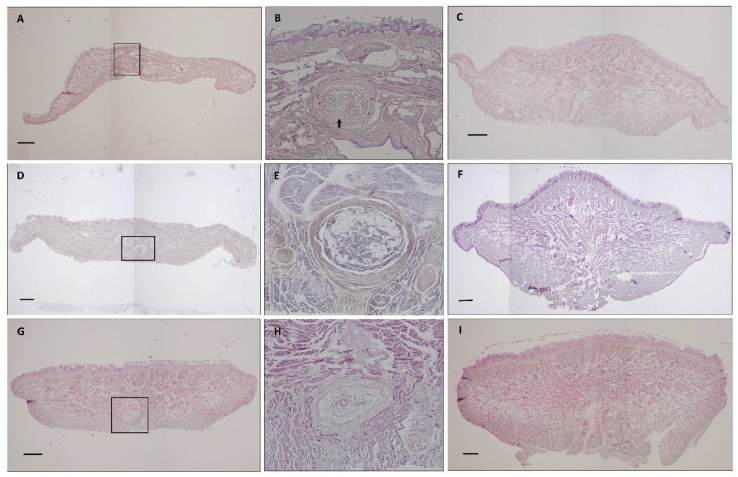
Photocomposition of a frontal section of the tongue’s body at the level of the anterior third (images **A**,**D**,**G**) or posterior third (images **C**,**F**,**I**) in *Lemur catta* (images **A**–**C**), *Varecia variegata* (images **D**–**F**) and *Eulemur macaco* (images **G**–**I**). Images (**B**,**D**,**H**) shown a magnification of the area framed in images (**A**,**D**,**G**), focusing on the encapsulated organ found in the antero-ventral position of the tongue in all species studied. Arrow in image (**B**) points to the presence of an cartilage only in *Lemur catta*. All sections were stained with hematoxylin—eosin. Scale bar 1 mm.

**Figure 5 animals-11-02811-f005:**
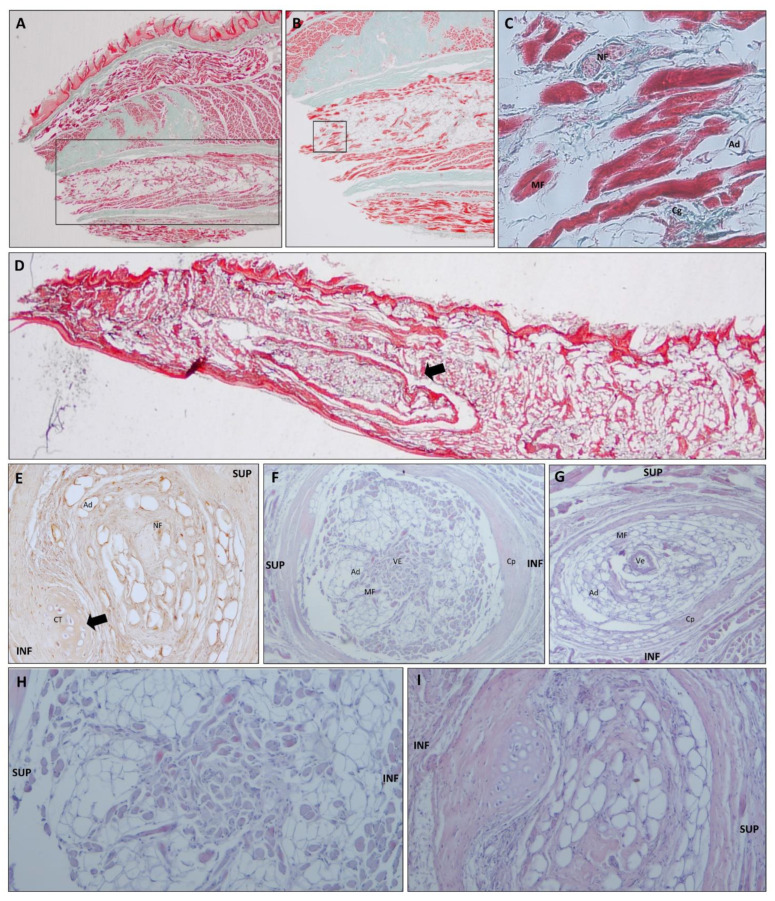
Images that illustrate the encapsulated structure present in the antero-ventral part of the tongue. Images (**A**–**C**,**E**,**I**) correspond to *Lemur catta*. Images (**D**,**F**,**H**) correspond to *Varecia variegata*. Image (**G**) correspond to *Eulemur macaco*. Images (**A**–**D**) (Masson trichrome stain) are from sagital-oriented longitudinal sections. Images (**E**) (S100 immunostain) and (**F**–**I**) (hematoxylin—eosin stain) are from frontal-oriented transversal sections. Some images, for descriptive purposes, have the following abbreviations: INF (inferior); SUP (superior); NF (nerve fiber); Ad (adipocyte); Cg (collagen); MF (muscle fiber); Ve (vascular element); CT (cartilaginous tissue) and Cp (capsule). In image (**A**) part of the encapsulated structure is remarked. Image (**C**) shows details of the tissue boxed in image (**B**). Image (**D**) shows a panoramic of the position of the encapsulated structure (arrow). Images captured with the following objectives: **A**, 10×; **B**, 20×; **C**, 40×; **D**, 4×; **E**–**G**, 10×; **H**,**I**, 20×.

**Figure 6 animals-11-02811-f006:**
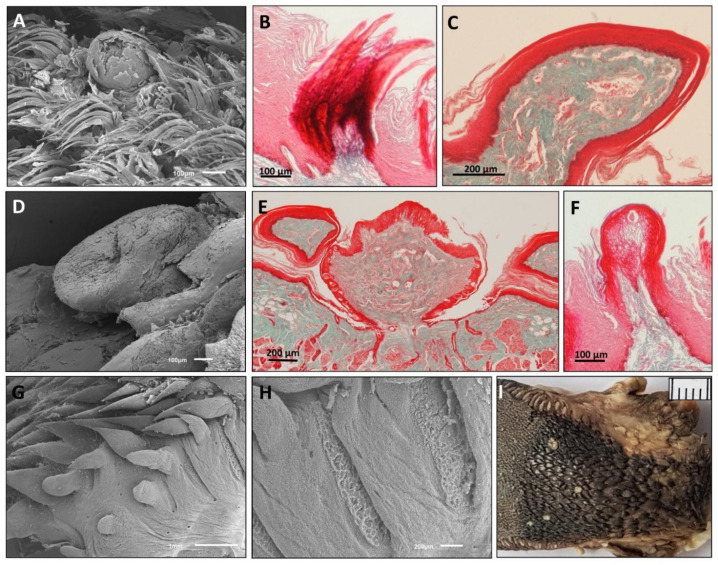
All images correspond to *Lemur catta* papillae. SEM images can be observed of fungiform and filiform (**A**), circumvallate and conical (**D**), and foliate and conical (**G**,**H**). Also presented are Masson trichrome-stained images of filiform (**B**), conical (**C**), circumvallate surrounded by conical (**E**) and a fungiform papilla displaying a taste bud at the top (**F**). Image (**I**) clearly shows the Y-shaped organization of circumvallate papillae. A width of one millimeter has the minor division in image (**I**).

**Figure 7 animals-11-02811-f007:**
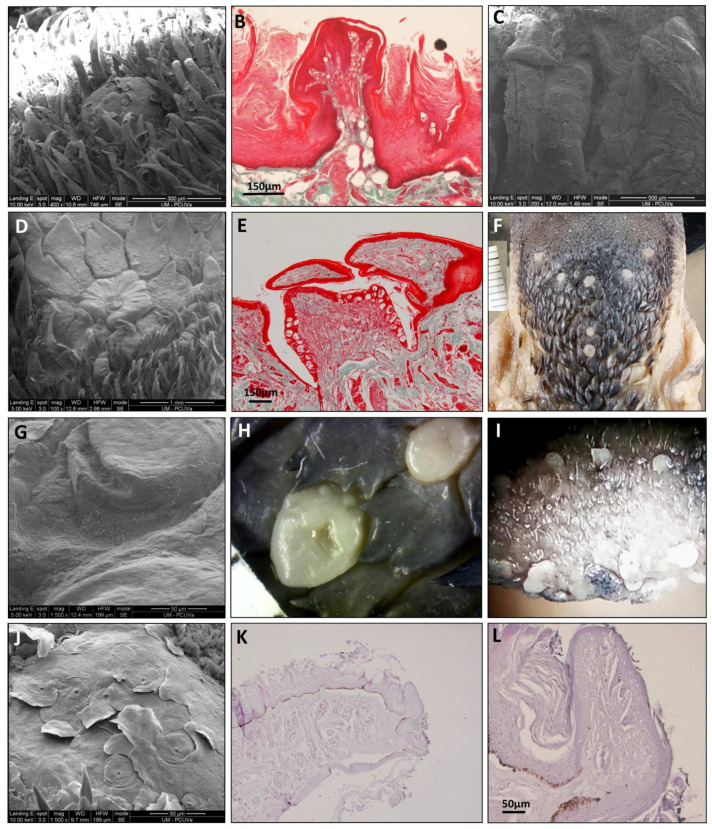
All images correspond to *Varecia variegata* papillae. SEM images can be observed of fungiform together with filiform (**A**), foliate (**C**), circumvallate (**D**,**F**), surrounded by flattened conical with several tips and some filiform in the inferior part of the image, a circumvallate central part (**G**, magnified from image **D** circumvallate papilla) and a superior view of hammerhead or maze-shaped papillae present on the tip of the tongue (**J**). Also presented are Masson trichrome-stained images of filiform and fungiform (**B**) and a circumvallate surrounded by conical (**E**). Images (**I**–**L**) illustrate the hammerhead or maze-shaped papillae present on the tip of the tongue, with K and L being sagittal sections stained with hematoxylin—eosin. Image (**F**) (a superior view of the root of the tongue) shows the Y-shaped disposition of circumvallate papillae and image H displays the crest shape of the flattened conical papillae that surround circumvallate ones in *Varecia variegata*. A scale of one millimeter in width, can be observed on image (**H**) (inferior left angle) and in image (**F**), the minor division is also one millimeter.

**Figure 8 animals-11-02811-f008:**
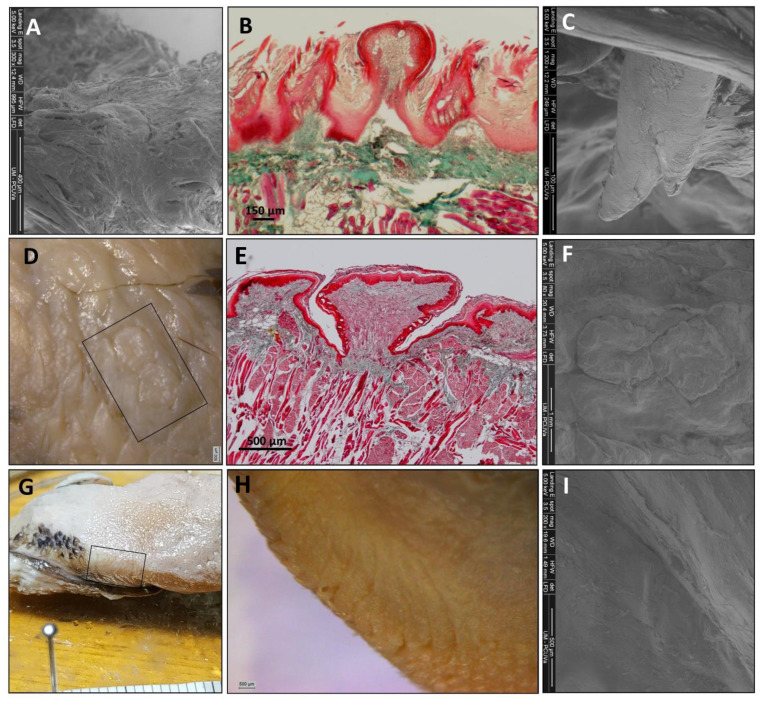
All images correspond to *Eulemur macaco* papillae. SEM images can be observed of fungiform and filiform (**A**), a conical-like papilla present on the tip of the tongue (**C**, magnified from Figure 3—image I right inferior angle), circumvallate and conical (**F**) and foliate (**I**). Also presented are Masson trichrome-stained images of filiform (**B**) and circumvallate (**E**) papillae. Optical images of circumvallate papillae (**D**, from the posterior branch of the Y-shaped disposition) and foliate papillae (**G**,**H**). Image (**D**,**F**) correspond to the same sample. The minor division of the scale in image (**G**) corresponds to one millimeter.

## Data Availability

Not applicable.

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
