# Peer review of "The Tongue in Three Species of Lemurs: Flower and Nectar Feeding Adaptations"

_animals, 2021, doi:10.3390/ani11102811_

Round 1

Reviewer 1 Report

In the introduction and discussion, it is necessary to include more current references.

In the material and methods, it is necessary to detail how the dissection and obtaining the tongues was carried out.

I don't think the term sublingua is appropriate. Did other authors endorse the term?

Lowercase letters appear in the images, capital letters appear in the caption! Correct.

The photomicrographs in figures 4 and 7 have low resolution quality. It is necessary to include new images with high resolution and calibration bar.

Author Response

In the introduction and discussion, it is necessary to include more current references.

The bibliography has been reviewed and missing bibliography added. Despite the fact that recent publications on the tongue and sublingua of the species object of the study are very scarce and the former descriptions are from 19th and 20th centuries, 22 articles published since 2000 have been included (one of them, translated from Japaness language) being 12 of the last 10 years (out of a total of 52 references). Taking as a reference the review published by Iwasaki in 2019 on the tongue of primates, all the references that he presents on lemurs in his review were cited in our study (Iwasaki et al., Comparative morphology of the primate tongue, Annals of Anatomy 2019 Volume 223, Pages 19-31). If the referee suggest us some reference important for the work we will add to them.

In the material and methods, it is necessary to detail how the dissection and obtaining the tongues was carried out.

The The description of the dissection of the sample, which was not included, has been added

I don't think the term sublingua is appropriate. Did other authors endorse the term?

This is the term used by the majority of the authors (see Iwasaky et al., 2019). There are several structures below the tongue:The sublingua, the plica sublingualis ( a fold of the oral mucosa covering the floor of the mouth and when is considered together with the caruncula as the frenal lamela, more over, Alouatta and Aotus have taste areas close to the caruncula and all together are called sublingual organ, but is different from sublingua, the New World monkeys do not present sublingua),  plica fimbriata, etc…

The following references describded and used the term sublingua:

Jones FW. 1918. The Sublingua and the Plica Fimbriata. J Anat. Jul;52(Pt 4):345-53.

Sonntag, C.F. 1920. The comparative anatomy of the tongues of the mammalian. I General description of the tongue. Proc. Zool. 507 Soc. Lond. IX: 115-129.

Sonntag, C.F. 1921. The comparative anatomy of the tongues of the mammalia. V. Family 5. Lemuroidea and Tarsioideae. Proc. 509 Zool. Soc. Lond. L: 741-754.

Hofer H O. 1989. Lightmicroscopical investigations of the sublingua of Microcebus murinus (Cheirogaleidae, Lemuriformes) with remarks on the phylogenetic relations of the tree shrews (Scandentia) to primates. Z Morphol Anthropol. 78(1):25-42.

Lowercase letters appear in the images, capital letters appear in the caption! Correct.

This mistake was corrected.

The photomicrographs in figures 4 and 7 have low resolution quality. It is necessary to include new images with high resolution and calibration bar.

The former images were draft images upload by a mistake. We replace all the images with better resolution plates and scale bars.

Reviewer 2 Report

I suggest that the present work has the analysis of its results revised and, consequently, adaptations in the discussions. The main suggestion is regarding the result in histology, the photographs in H&E are in magnification that it is not possible to observe tissue details and morphological recognition of structures that could be further explored and described with richness. There is no general histological description or specific structure of the encapsulated organ, it is not possible to recognize its tissue elements. Below I will leave some examples and questions:

1. The photograph 4 describes the presence of cartilage, but it is not possible to identify it. When zoomed in the image loses sharpness. I'm not sure about the description of this structure or encapsulated organ found. I believe a close-up photograph (with larger lenses). Histological photographs could be more detailed and rich in detail. The photomicrographs shown in Figure 4 do not add much information, the photographs could have been taken with a zoom (40x objective) to enrich the details. In the displayed magnification, it is not possible to identify tissue details, not even in the higher magnification photos shown in B, E and H focusing on the encapsulated.

2. I suggest that the encapsulated organ should be described according to its histological characteristics, as photography does not allow us to identify the tissues and details present in this structure.

3. In figure 5 what purpose was the Masson trichrome stained technique used (to differentiate connective tissue fibers and muscle fibers? Would H&E staining be more indicated to demonstrate details of the epithelium (including taste buds)? I believe that histology was very little explored in this work and could enrich with morphological details.

Author Response

I suggest that the present work has the analysis of its results revised and, consequently, adaptations in the discussions. The main suggestion is regarding the result in histology, the photographs in H&E are in magnification that it is not possible to observe tissue details and morphological recognition of structures that could be further explored and described with richness. There is no general histological description or specific structure of the encapsulated organ, it is not possible to recognize its tissue elements. Below I will leave some examples and questions:

New references were added.

  1. The photograph 4 describes the presence of cartilage, but it is not possible to identify it. When zoomed in the image loses sharpness. I'm not sure about the description of this structure or encapsulated organ found. I believe a close-up photograph (with larger lenses). Histological photographs could be more detailed and rich in detail. The photomicrographs shown in Figure 4 do not add much information, the photographs could have been taken with a zoom (40x objective) to enrich the details. In the displayed magnification, it is not possible to identify tissue details, not even in the higher magnification photos shown in B, E and H focusing on the encapsulated.

A new figure (figure 8) with images of the encapsulated structure has been included. In these images different elements of the tissues have been marked with abbreviations, in order to clarify the description. An image with an immunohistochemical staining has been included, marking adipocytes, nerve fibers and cartilage. Low resolution images have been replaced.

  1. I suggest that the encapsulated organ should be described according to its histological characteristics, as photography does not allow us to identify the tissues and details present in this structure.

The description in the text has been revised and in the images different elements of the tissues have been marked with abbreviations, in order to clarify the description.

  1. In figure 5 what purpose was the Masson trichrome stained technique used (to differentiate connective tissue fibers and muscle fibers? Would H&E staining be more indicated to demonstrate details of the epithelium (including taste buds)? I believe that histology was very little explored in this work and could enrich with morphological details.

These two stains are to accompany the images of the morphology of the papillae. The idea of the work is to show morphological adaptations to the way of feeding without delving into the histology that will be addressed in future works if we get better samples. These samples are quite hardened because they come from a collection with pieces that have been submerged in formaldehyde for more than 10 years, that come from dead animals frozen for transport and that have not been perfused or treated in the best conditions for histology. On the other hand, in this work it is requested to be concise and adjust to the theme of the journal, and an adequate analysis of histological details such as taste buds (which we will try to see with semi-fine sections of resins and with TEM in future works) and other histological aspects would exceed the length of a typical "Animals" article

Round 2

Reviewer 1 Report

Photomicrographs 4 are still of low quality

Author Response

Photomicrographs 4 are still of low quality

  • We remake figure 4 for improve the resolution as much as we can. This plate was designed to have a global perspective of the section of the tongue so we had to use 1x objective (with less resolution). Using other objectives the panoramic needs too many photos and the final plate seems like a collage showing numerous lines owing to the superposition of the individual pictures.
  • We also review the English language (task done by our co-author born in the USA). 
